

# FUSTr: a tool to find gene families under selection in transcriptomes

T. Jeffrey Cole and Michael S. Brewer

Department of Biology, East Carolina University, Greenville, NC, United States of America

## ABSTRACT

**Background.** The recent proliferation of large amounts of biodiversity transcriptomic data has resulted in an ever-expanding need for scalable and user-friendly tools capable of answering large scale molecular evolution questions. FUSTr identifies gene families involved in the process of adaptation. This is a tool that finds genes in transcriptomic datasets under strong positive selection that automatically detects isoform designation patterns in transcriptome assemblies to maximize phylogenetic independence in downstream analysis.

**Results.** When applied to previously studied spider transcriptomic data as well as simulated data, FUSTr successfully grouped coding sequences into proper gene families as well as correctly identified those under strong positive selection in relatively little time.

**Conclusions.** FUSTr provides a useful tool for novice bioinformaticians to characterize the molecular evolution of organisms throughout the tree of life using large transcriptomic biodiversity datasets and can utilize multi-processor high-performance computational facilities.

Corresponding authors
T. Jeffrey Cole,
coleti16@students.ecu.edu
Michael S. Brewer,
brewermi14@ecu.edu

# BACKGROUND

Elucidating patterns and processes involved in the adaptive evolution of genes and genomes of organisms is fundamental to understanding the vast phenotypic diversity found in nature. Recent advances in RNA-Seq technologies have played a pivotal role in expanding knowledge of molecular evolution through the generation of an abundance of protein coding sequence data across all levels of biodiversity (*Todd, Black & Gemmell, 2016*). In non-model eukaryotic systems, transcriptomic experiments have become the *de facto* approach for functional genomics in lieu of whole genome sequencing. This is due largely to lower costs, better targeting of coding sequences, and enhanced exploration of post-transcriptional modifications and differential gene expression (*Wang, Gerstein & Snyder, 2009*). This influx of transcriptomic data has resulted in a need for scalable tools capable of elucidating broad evolutionary patterns in large biodiversity datasets.

Billions of years of evolutionary processes gave rise to remarkably complex genomic architectures across the tree of life. Numerous speciation events along with frequent whole genome duplications have given rise to myriad multigene families with varying roles in the processes of adaptation (*Benton, 2015*). Grouping protein encoding genes

into their respective families *de novo* has remained a difficult task computationally. This typically entails homology searches in large amino acid sequence similarity networks with graph partitioning algorithms to cluster coding sequences into transitive groups (*Andreev & Racke, 2006*). This is further complicated in eukaryotic transcriptome datasets that contain several isoforms via alternative splicing (*Matlin, Clark & Smith, 2005*). Further exploration of Darwinian positive selection in these families is also nontrivial, requiring robust Maximum Likelihood and Bayesian phylogenetic approaches.

Here we present a fast tool for finding Families Under Selection in Transcriptomes (FUSTr), to address the difficulties of characterizing molecular evolution in large-scale transcriptomic datasets. FUSTr can be used to classify selective regimes on homologous groups of phylogenetically independent coding sequences in transcriptomic datasets and has been verified using large transcriptomic datasets and simulated datasets. The presented pipeline implements a simplified user experience with minimized third-party dependencies, in an environment robust to breaking changes to maximize long-term reproducibility.

While FUSTr fills a novel niche among sequence evolution pipeline, a recent tool, VESPA (*Webb, Thomas & Mary, 2017*), performs several similar functions. Our tool differs in that it can accept *de novo* transcriptome assemblies that are not predicted ORFs. VESPA requires nucleotide data to be in complete coding frames and does not filter isoforms or utilize transitive clustering to deal with domain chaining. Additionally, VESPA makes use of slow maximum likelihood methods for tests of selection and provides no information about purifying selection, whereas FUSTr utilizes a Fast Unconstrained Bayesian Approximation (FUBAR) (*Murrell et al., 2013*) to analyze both pervasive and purifying regimes of selection.

## IMPLEMENTATION

FUSTr is written in Python with all data filtration, preparation steps, and command line arguments/parameters for external programs contained in the workflow engine Snakemake (*Köster & Rahmann, 2012*). Snakemake allows FUSTr to operate on high performance computational facilities, while also maintaining ease of reproducibility. FUSTr and all third-party dependencies are distributed as a Docker container (*Merkel, 2014*). FUSTr contains ten subroutines that takes transcriptome assembly FASTA formatted files from any number of taxa as input and infers gene families that are either under diversifying or purifying selection. A graphical overview of this workflow and parallelization scheme has been outlined in Fig. 1.

*Data Preprocessing.* The first subroutine of FUSTr acts as a quality check step to ensure input files are in valid FASTA format. Spurious special characters resulting from transferring text files between multiple operating system architectures are detected and removed to facilitate downstream analysis.

*Isoform detection.* Header patterns are analyzed to auto-detect whether the given assembly includes isoforms by detecting naming convention redundancies commonly used in isoform designations, in addition to comparing the header patterns to common assemblers such as Trinity *de novo* assemblies (*Haas et al., 2013*) and Cufflinks reference genome guided assemblies (*Trapnell et al., 2014*).

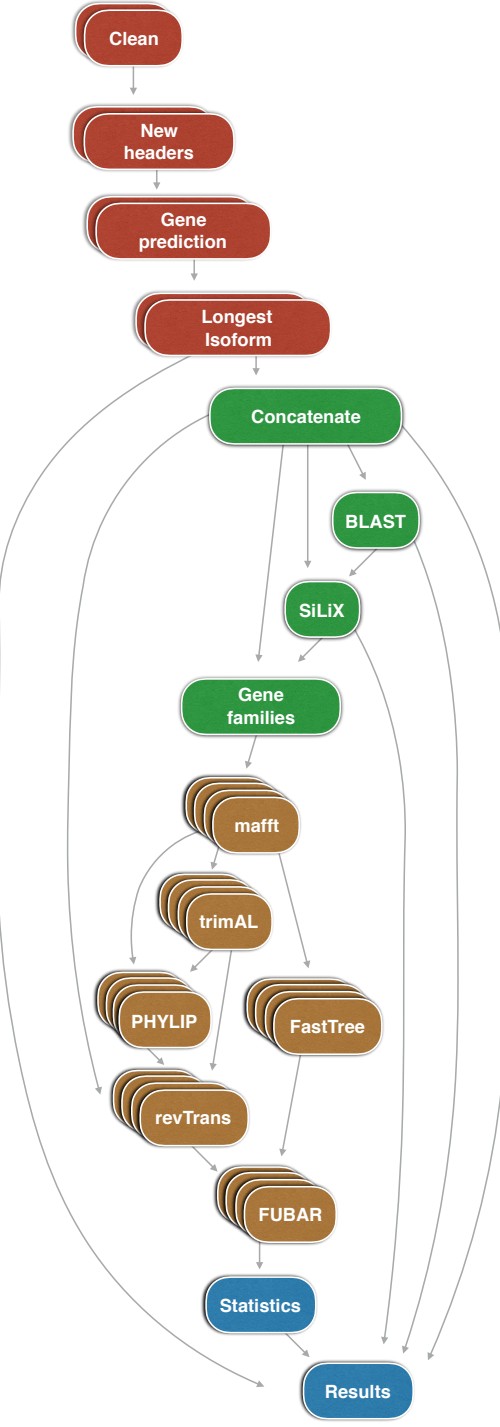

**Figure 1** **Parallelization scheme and workflow of FUSTr.** Color coding denotes functional subroutines in the pipeline: preparation and open reading frame prediction (red); homology inferenece and gene family clustering (green); multiple sequence alignment, phylogenetics, and selection detection (brown); and model selection and reconciliation (blue).

*Gene prediction*. Coding sequences are extracted from transcripts using Transdecoder v3.0.1 (*Haas et al., 2013*). Transdecoder predicts Open Reading Frames (ORFs) using likelihood-based approaches. A single best ORF for each transcript with predicted coding sequence is extracted providing nucleotide coding sequences (CDS) and complementary amino acid sequences. This facilitates further analyses requiring codon level sequences while using the more informative amino acid sequences for homology inferences and multiple sequence alignments. If the data contain several isoforms of the same gene, at this point only the longest isoform is kept for further analysis to ensure phylogenetic independence. The user may customize the use of TransDecoder by changing minimum coding sequence length (default: 30 codons) or strand-specificity (default: off). Users also have the option to only retain ORFs with homology to known proteins through a BLAST search against Uniref90 or Swissprot in addition to searching PFAM to identify common protein domains.

*Homology search*. The remaining coding sequences are assigned a unique identifier and then concatenated into one FASTA file. Homologies among peptide sequences are assessed via BLASTP acceleration through DIAMOND (v.0.9.10) with an *e*-value cutoff of $10^{-5}$.

*Gene Family inference*. The resulting homology network is parsed into putative gene families using transitive clustering with SiLiX v.1.2.11, which is faster and has better memory allocation than other clustering algorithms such as MCL and greatly reduces the problem of domain chaining (*Miele, Penel & Duret, 2011*). Sequences are only added to a family with 35% minimum identity, 90% minimum overlap, with minimum length to accept partial sequences in families as 100 amino acids, and minimum overlap to accept partial sequences of 50%. These are the optimal configurations of SiLiX (*Bernardes et al., 2015*), but the user is free to configure these options.

*Multiple sequence alignment and phylogenetic reconstruction*. Multiple amino acid sequence alignments of each family are then generated using the appropriate algorithm automatically detected using MAFFT v7.221 (*Katoh & Standley, 2013*). Spurious columns in alignments are removed with Trimal v1.4.1's *gappyout* algorithm (*Capella-Gutiérrez & Silla-Martínez, 2009*). Phylogenies of each family's untrimmed amino acid multiple sequence alignment are reconstructed using FastTree v2.1.9 (*Price, Dehal & Arkin, 2010*). Trimmed multiple sequence codon alignments are then generated by reverse translation of the amino acid alignment using the CDS sequences.

*Tests for selective regimes*. Families containing at least 15 sequences have the necessary statistical power for tests of adaptive evolution (*Wong et al., 2004*). Tests of pervasive positive selection at site specific amino acid level are implemented with FUBAR (*Murrell et al., 2013*). Unlike codeml, FUBAR allows for tests of both positive and negative selection using an ultra-fast Markov chain Monte Carlo routine that averages over numerous predefined site-classes. When compared to codeml, FUBAR performs as much as 100 times faster (*Murrell et al., 2013*). Default settings for FUBAR, as used in FUSTr, include twenty grid points per dimension, five chains of length 2,000,000 (with the first 1,000,000 discarded as burn-in), 100 samples drawn from each chain, and concentration parameter of the Dirichlet prior set to 0.5.

Users have the option to also run tests for pervasive selection using the much slower CODEML v4.9 (*Yang, 2007*) with the codon alignments and inferred phylogenies. Log-likelihood values of codon substitution models that allow positive selection are then compared to respective nested models not allowing positive selection (M0/M3, M1a/M2a, M7/M8, M8a/M8); Bayes Empirical Bayes (BEB) analysis then determines posterior probabilities that the ratio of nonsynonymous to synonymous substitutions ($d_N/d_S$) exceeds one for individual amino acid sites.

*Final output and results.* The final output is a summary file describing which gene families were detected and those that are under strong selection and the average $d_N/d_S$ per family. A CSV file for each family under selection is generated giving the following details per codon position of the family alignment: alpha mean posterior synonymous substitution rate at a site; beta mean posterior non-synonymous substitution rate at a site; mean posterior beta-alpha; posterior probability of negative selection at a site; posterior probability of positive selection at a site; Empiricial Bayes Factor for positive selection at a site; potential scale reduction factor; and estimated effective sample site for the probability that beta exceeds alpha.

## Validation

We tested FUSTr on six published whole body transcriptome sequences from an adaptive radiation of Hawaiian *Tetragnatha* spiders (NCBI Short Read Archive accession numbers: SRX612486, SRX612485, SRX612477, SRX612466, SRX559940, SRX559918) assembled using the same methods as the original publication (*Brewer et al., 2015*). Spider genomes contain numerous gene duplications lending to gene family rich transcriptomes. Additionally, this adaptive radiation has been shown to facilitate strong, positive, sequence-level selection in these transcriptomes (*Brewer et al., 2015*). This dataset provides an ideal case use for FUSTr.

A total of 273,221 transcripts from all six *Tetragnatha* samples were provided as input for FUSTr, and a total of 4,258 isoforms were removed leaving 159,464 coding sequences for analysis after gene prediction. The entire analysis ran in 13.7 core hours, completing within an hour when executed on a 24-core server. Time to completion and memory usage for each of FUSTr's subroutines performance in this analysis is reported in Table 1. FUSTr recovered 134 families containing at least 15 sequences. Of these 46 families contained sites under pervasive positive selection while all families also contained sites under strong purifying selection. This can be contrasted with the analysis by *Brewer et al. (2015)*, which found 2,647 one-to-one six-member orthologous loci (one ortholog per each of the same samples), with 65 loci receiving positive selection based on branch-specific analysis. The original analysis did not allow paralogs whereas FUSTr does not reconstruct one-to-one orthogroups but entire putative gene families, and the selection analysis utilized by FUSTr is site-specific and not branch-specific. Thus, it is not expected that the results from FUSTr would perfectly match up with the original analysis; however, five of the 46 families FUSTr found to be under selection included loci from *Brewer et al.*'s (*2015*) original 65 under selection based on branch-specific analysis.
**Table 1** **Benchmarks for each subroutines' time and memory used for the *Tetragnatha* transcriptome assembly analysis.** Red highlighted row represents subroutine consuming the most memory and time per task, blue highlighted row represents subroutine consuming the most memory and time in total.

| Subroutine | Tasks | Seconds per task | Total seconds | RAM per task (MiB) | Total RAM (MiB) |
|---|---|---|---|---|---|
| Clean fastas | 6 | 1.40 | 8.38 | 46.5 | 278.9 |
| New headers | 6 | 1.65 | 9.90 | 43.6 | 261.5 |
| Long isoform | 6 | 0.512 | 3.07 | 51.5 | 309.13 |
| Transdecoder | 1 | 10,436.7 | 10,436.7 | 3,249.8 | 3,249.8 |
| Diamond | 1 | 32.1 | 32.1 | 234.0 | 234.0 |
| SiLiX | 1 | 4.51 | 4.51 | 22.8 | 22.8 |
| Mafft | 135 | 3.24 | 437.8 | 18.3 | 2,466.5 |
| FastTree | 135 | 3.09 | 417.4 | 18.5 | 2,491.3 |
| TrimAL | 135 | 1.87 | 252.2 | 17.9 | 2,415.6 |
| FUBAR | 135 | 278.6 | 37,605.5 | 28.8 | 3,886.2 |

The same 273,221 transcripts were entered as input for VESPA as a comparative analysis. Because VESPA cannot detect and filter ORFs in transcripts, it was unable to infer proper coding sequences. In its first phase of cleaning input FASTA files, 86,269 transcripts were wrongly removed for having "internal stop codons" via improper reading frame inference, and 182,000 transcripts were removed due to "abnormal sequence length." Approximately 98% of the transcripts were removed in the first phase of VESPA with no gene predictions, rendering further analysis unnecessary for proper comparison of the performance of the two pipelines.

We further validated FUSTr using coding sequences from simulated gene families with predetermined selective regimes. We used EvolveAGene (*Hall, 2007*) on 3,000 random coding sequences of a random length of 300–500 codons to generate gene families containing 16 sequences evolved along a symmetric phylogeny each with average branch lengths chosen randomly between 0.01–0.20 evolutionary units. Selective regimes with a selection modifier of 3.0 were randomly chosen for each family so that a random 10% partition of the family received pervasive positive selection, purifying selection, or constant selection. All other settings for EvolveAGene were left as their defaults: the probability of accepting an insertion = 0.1, the probability of accepting a deletion = 0.025, the probability of accepting a replacement = 0.016, and no recombination was allowed. A visual schema for these simulations can be found in Fig. 2.

The resulting 48,000 simulated sequences were used as input for FUSTr with TransDecoder set to be strand-specific. FUSTr correctly recovered all 3,000 families, and all 975 that were randomly selected to undergo strong positive selection were correctly classified as receiving pervasive positive selection. Additionally, the families selected to undergo purifying selection were correctly classified, and families selected to receive constant selection were classified as not having any specific sites undergoing purifying or pervasive positive selection. Scripts for these simulations can be found at https://github.com/tijeco/FUSTr.

Cole and Brewer (2018), *PeerJ*, DOI 10.7717/peerj.4234

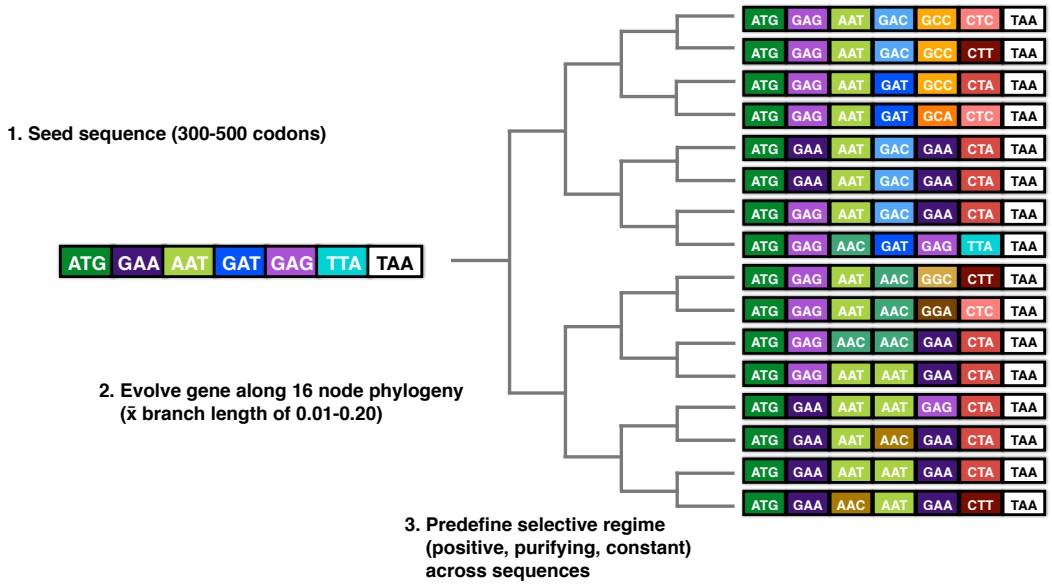

**Figure 2 Schematic of EvolveAGene methods used to simulate sequences for the validation of FUSTr.** Sequences were randomly generated and evolved along a symmetric phylogeny under a given selective regime (positive, negative, or constant across the entire gene).

# CONCLUSIONS

Current advances in RNA-seq technologies have allowed for a rapid proliferation of transcriptomic datasets in numerous non-model study systems. It is currently the only tool equipped to deal with the nuances of transcriptomic data, allowing for proper prediction of gene sequences and isoform filtration. FUSTr provides a fast and useful tool for novice bioinformaticians to detect gene families in transcriptomes under strong selection. Results from this tool can provide information about candidate genes involved in the processes of adaptation, in addition to contributing to functional genome annotation.

# ACKNOWLEDGEMENTS

This work would not have been possible without XSEDE computational allocations (BIO160060). We also thank Chris Cohen for editing this manuscript.

## Funding

This work was supported by National Science Foundation Graduate Research Fellowship and the East Carolina University Department of Biology. The funders had no role in study design, data collection and analysis, decision to publish, or preparation of the manuscript.

## Grant Disclosures

The following grant information was disclosed by the authors:
National Science Foundation Graduate Research Fellowship.
East Carolina University Department of Biology.

## Competing Interests

The authors declare there are no competing interests.

## Author Contributions

- T. Jeffrey Cole conceived and designed the experiments, performed the experiments, analyzed the data, wrote the paper, prepared figures and/or tables, reviewed drafts of the paper.
- Michael S. Brewer conceived and designed the experiments, performed the experiments, analyzed the data, contributed reagents/materials/analysis tools, wrote the paper, reviewed drafts of the paper.

## Data Availability

Github: https://github.com/tijeco/FUSTr.

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
