# Peer review of "FUSTr: a tool to find gene families under selection in transcriptomes"

_PeerJ, doi:10.7717/peerj.4234_

## Round 0.1 · original submission · Major Revisions

Please address the comments raised by the reviewers. If needed, Reviewer 2 will be happy to share the files of his test.

Reviewer 1 ·

Basic reporting

In figure 2, a clearer description of the bar plot would be helpful. It is not very clear whether the x-axis represents the mean dN/dS of each gene family. If yes, the authors may also consider to add error bars.

Experimental design

1. In the validation section, the authors may consider to include the performance comparison with other existing tools such as VESPA.
2. In the validation section, 2nd paragraph, the authors show that their method has very high sensitivity by identifying all 50 families that were simulated to undergo pervasive positive selection. The authors may consider to add genes that don’t undergo positive selection into their simulated data. It could be helpful to demonstrate specificity of their method.

Validity of the findings

No comment.

Comments for the author

The authors developed a useful tool to detect the gene families and amino acid sites under positive selection based on transcriptomic data. This tool differs from other existing methods by allowing de novo transcriptome assemblies that are not predicted ORFs. The automated tool also simplifies the installation and analysis process. I think this paper will be of interest to the readership of PeerJ.

·

Basic reporting

Besides the inconsistent "raw data", this report fits PeerJ's other "basic reporting" requirements. It must be stressed that while the dataset used for validation is very problematic (see "Problems with the validation dataset" in the "general comments"), the main issue here is that it is not adequate to test the performance of the pipeline reported here (see "Main issues" in the "general comments") and thus the pipeline should be tested using a different dataset.

Experimental design

Methods need to be described in further detail. In particular authors need to provide a more thorough documentation in the git repository (e.g. describe each subroutine and the intermediate files created by the pipeline) and a brief outline of the underlying procedures in the report for publication. Take advantage of the online-only nature of PeerJ and be as descriptive as possible. Remind that the better your documentation is and the easier for the user to understand how the pipeline is handling the data, the more users you'll have.

I celebrate that the authors have made available their method, so I was able to reproduce their analysis. For my testing I generate a non-redundant dataset of the sequences in Supplemental table 1 (530, removing 53 that are duplicates) and launched the FUSTr in a Linux machine with 4 processors. I labelled each sequence with its corresponding accession number, a short code for the species, and the "family classification" from the original publication (there were 9 "families" in the original publication).

>{gb ID}|{spe}|{family}

I then generated one fasta file per species (trying to emulate different transcriptomes for different "samples") and named the files as the short species codes. The 49 files contain between 192 and 1 sequences (there are 18 species for which contributing with only one sequence to the dataset). Trying to reproduce the potentially conflictive redundancy of annotation of the original publication and test the ability of FUSTr to recognize header patterns, I left four duplicated sequences labelled as both "ICKs" and "Huwentoxin-1_family" (i.e. they share the first two fields of the header). The input fasta files thus contain a total of 534 sequences, with four duplicates.

The pipeline is encoded in a Snakefile , for which I tracked back the calls and subroutines. The first subroutine is to duplicate the fasta files and stored them in "{sample}.clean" files, then headers are changed to comply with the conventions for the next steps. At this point, headers patterns are extracted from the files by the rule "newHeaders" and stored in <headerPatterns.txt> file. Trinity's Transdecoder.LongOrfs is next called with a predefined "-m 30" option (minimal length for an ORF), which generates as much as six potential ORFs (the longest per frame), and the protein and nucleotide sequences are extracted with TransDecoder.Predict run on the <longest_orfs.pep> file, which predicts a single "best" ORF per input sequence. An important control for validation at this point would be to check if the predicted protein sequences correspond indeed to the known translated sequences (a quick check at UniProt reveals that there are 133 transcripts in the dataset for which there is evidence at protein level). In my test, these steps removed 12 sequences from the analysis for no clear reasons as inspecting the Transdecoder intermediate files reveals that for all of them an ORF containing all of the known coding sequence was detected. The pipeline then assign a specific short ID for each best transcript model and writes two fasta files with all sequences as amino acids (pep) and nucleotides (cds).

An all-vs-all blastp is then run on the <all.pep.combined> file with a predefined -evalue 1E-5 and the output piped into SiLiX for clustering with a predefined minimum overlap of 0.9 (-r option). Per "family" fasta files are then generated by the node2families subroutine and large enough families (>14 sequences, it would be good if the authors provide a rationale for this choose) aligned with MAFFT, trimmed with trimAl and transformed into phylip (so codeml can be run). The protein alignments are also use to compute gene trees with FastTree. For a dataset as the one used for validation, all the previous steps are essentially completed in no time (less than a minute in my Linux with four 2.93GHz cores). I didn't have time to systematically check all retained ORFs, but in at least one "family" containing sequences named "lycotoxin" in the original publication (corresponding to IDs labelled as families 10 and 11 in the Supplemental Table 1) the predicted ORFs do not correspond to the known translated sequences.

For instance, the "best" ORF of FM863950 is

>Gene.187::FM863950_Lycsi_U1-lycotoxin_family::g.187::m.187 Gene.187::FM863950_Lycsi_U1-lycotoxin_family::g.187 ORF type:5prime_partial len:47 (-) FM863950_Lycsi_U1-lycotoxin_family:459-599(-)
TTTTTTTTTTTTTTTTTTTTTTTTTTTTTTTTTTTTTTTTGGAATCCTCAGAACTTTATCCAATTTAAAGAATGTTACAAACAAAAAAAATCAAAACAATTTTCTATGTCATTTTTGCTATCGTTTTACGAAATAATTTAA

Translated as

>Gene.187::FM863950_Lycsi_U1-lycotoxin_family::g.187::m.187 Gene.187::FM863950_Lycsi_U1-lycotoxin_family::g.187 ORF type:5prime_partial len:47 (-) FM863950_Lycsi_U1-lycotoxin_family:459-599(-)
FFFFFFFFFFFFFWNPQNFIQFKECYKQKKSKQFSMSFLLSFYEII*

But the original sequence is

>FM863950.1 Lycosa singoriensis mRNA for toxin-like structure LSTX-A23 precursor (LSTX-A23 gene)
GACAGAACTTCGGTTTAGTTCCCCAGGAAATTTTGACTAAGTGACATCTTGAGGTTTCCTCCCAGCCAATCATGATGAAGGTTCTAGTGGTCGTTGCTCTTTTGGTTACTCTTATCAGTTACTCTTCAAGTGAAGGGATTGACGATCTTGAAGCTGACGAACTGTTGTCTTTAATGGCCAACGAGCAAACCAGGAAAGAATGCATTCCCAAACACCACGAATGTACGAGCAATAAGCACGGCTGCTGTAGGGGTAACTTTTTCAAATACAAATGCCAGTGTACAACAGTTGTTACCCAGGACGGAGAACAGACCGAAAGATGCTTCTGTGGAACTCCCCCTCACCACAAGGCGGCCGAATTGGTGGTTGGCTTCGGGAGGAAGATTTTCGGATAAAAGAACAGCTTTATCGGAATATGTGAAGACACGTTATTTGACGTAAATGAACCTCTGTAGAGTTTAAATTATTTCGTAAAACGATAGCAAAAATGACATAGAAAATTGTTTTGATTTTTTTTGTTTGTAACATTCTTTAAATTGGATAAAGTTCTGAGGATTCCAAAAAAAAAAAAAAAAAAAAAAAAAAAAAAAAAAAAAAAAA

Translated as
>CAS03548.1 toxin-like structure LSTX-A23 precursor [Lycosa singoriensis]
MMKVLVVVALLVTLISYSSSEGIDDLEADELLSLMANEQTRKECIPKHHECTSNKHGCCRGNFFKYKCQCTTVVTQDGEQTERCFCGTPPHHKAAELVVGFGRKIFG

Thus, the identification of the coding regions by Transdecoder on this dataset is not optimal (notice that the predicted ORF contains the polyA region of the original sequence). This means that the gene tree and the selection tests were not performed with the coding sequences, and the results are therefore bogus. The authors must debug this problem and document their findings (e.g. is there a way to optimize the selection of the best ORF?).

Moving on, the last and most computationally expensive step performs codeml analysis (using a biopython-inspired wrapper) under 7 different models of dN/dS distributions and testing for the most likely model of selection (achieved by comparing the log likelihoods). In my test, 9 families were automatically processed in just under 20 hours. If I understand correctly the summary file (a three-column file not described in the report, nor in the git, with four model comparisons per family), in my test three families are under "positive selection" (i.e. p<=1e-2 for models allowing dN/dS>1 vs models with fixed or variable dN/dS<=1), two of which correspond to kunitz toxins, in-line with what is reported by the authors, but also including the family containing the bogus ORFs of lycotoxins. Although the authors have not documented how they obtained the plot in Figure 2 (alas, it would be better if they do), I parsed the *mcl files and found 15 and 5 sites with dn/ds>1 in each of the kunitz families (I suppose corresponding to "kunitz 1" and "kunitz 2", respectively, in the figure).

Validity of the findings

The "validation" and simulated datasets are not adequate to test the performance of FUSTr. The amount of data in both cases is unrealistically small for a typical transcriptome study, let alone for a tool intended to analyze several transcriptomes at once.

Comments for the author

Suggestions for improvement
- Perform the "validation" on a relevant dataset, this should include potentially conflicting sequences (e.g. multiple isoforms derived from a single gene) and identifiers. One alternative would be to generate Trinity/cufflinks transcriptomes from publicly available SRAs known to be rich on multi-gene families
- Document all the steps performed by the pipeline (the more detailed the better)
- Include the raw fasta files for the validation datasets in the git repository
- If authors choose to maintain the current "validation" dataset as an additional proof of the performance, the dataset must be corrected for the several inconsistencies detailed below and the inconsistencies in the original dataset described
- Provide the results of the simulated dataset in graphical form

Main issues
1. "Validation" dataset is not adequate for testing the features of the pipeline
The dataset used is a manually curated set of full precursor sequences originally analyzed in PLoS Genetics 11:e1005596 and contains less than 600 sequences. This is an unrealistic size for transcriptome based studies, where thousands of transcripts would be predicted from a single experiment.

2. Insufficient description of the methods
The pipeline is not properly documented, with several steps not even described (e.g. the construction of new headers and the processing of the codeml outputs), non described third-party options (e.g. blastp evalue and SiLiX minimum overlap). See extensive comments in the "Experimental Design" section.


Problems with the "validation" dataset
I will comment on the several inconsistencies of what I consider very problematic "raw data" (in this case the dataset used for the validation part), but it should be kept in mind that, as described above, the dataset used for validation is not adequate for testing the purposed use of FUSTr.

- The number of sequences analyzed here is reported to be 624 in the text (marks 100-101), but numbers in Table 1 add up to only 618, whereas there are only 583 accession numbers in the supplementary file. I would have expected the authors to identify this not minor inconsistency before, but I understand it may be related to the not well structured presentation of the original publication (PLoS Genetics 11:e1005596, 2015). For the sake of this review, I have downloaded the relevant files from https://doi.org/10.1371/journal.pgen.1005596.s005 (including the reportedly accession numbers reportedly used) and https://doi.org/10.1371/journal.pgen.1005596.s006 (reportedly including the codon based alignments used for testing selection). Whereas the total number of analyzed sequences is not reported in the original publication, there are nine named "spider toxin families" in the supplementary files (alas somehow ten are plotted in figure 1of that publication), including 708 sequences in the alignments (four sequences appear in two different alignment files) and only 686 in the accession number tables. I regret these inconsistencies were left unattended in the original paper, but here the authors have the chance to correct the public record. I could map 567 identifiers shared between this report and the original publication, the remaining 16 identifiers correspond to the "FUSTr family 5" (omega hexatoxin in Supplemental Figure 4) for which the IDs in the original publication were not-standard.

- The number of "highly curated spider venom toxin protein families" analyzed is said to be seven in the text (mark 100), all of which are said to be correctly recovered by FUSTr (mark 103). The eight bars in Figure 2 reflect the fact that "kunitz toxins" were split into two groups by SiLiX's step of FUSTr). Further inspection of the Supplemental Table 1, reveals that the 583 sequences are actually classified into 23 similarity clusters by the pipeline, nine of which contain 10 or more sequences and including five cases in which several FUSTr-derived clusters mapping to a single "family" as (ill-) defined in the original publication.

- The dataset analyzed also has problems regarding the assumption of phylogenetic independence (identical sequences, unknown biological sources, erratic phylogenetic breadth, etc).

Reviewer 3 ·

Basic reporting

no comment

Experimental design

It is not very clear to me how the simulated data generated in validation section (Starting from line 106). Also, in the implementation part, it seems that FUSTr performs many processing steps for input data. However, one question I have is that is this a data-driven model? For example, does the output purely come from the clustering results? Is FUSTr aimed to identify how many gene families exist in the input sequence?

Validity of the findings

no comment

---

## Round 0.2 · Minor Revisions

Please address the additional comments from the reviewer.

·

Basic reporting

I agree that the new real life dataset provides an "ideal case use for FUSTr", however is not really a benchmark. In the original publication, more than 2000 homologous loci were identified in six specimens from two species. In that publication, less than 70 loci were identified as being under selection for color-changing or colonization phenotypes. I think it would be interesting if the authors double check if the families identified as under selection by FUSTr are the same. It would be also interesting to see if other multi gene families are also under selection.

Experimental design

Although I'm happy with authors responses to the previous round of revisions, in particular the with the new validation dataset, and that I celebrate the almost complete re-engineering of the pipeline, which now relies on Docker as container and snakemake for workflow management, I would encourage the authors to further document the potential problems their intended users could face. For instance, Docker requires root privileges for running commands or to create a group of users that also needs root privileges, which many users might not have. Also is not clear for me if the setup_docker should be run every single FUSTr run (i.e. with different data folders). This time I was not able to reproduce authors' results, due to conflicts with my system configuration. Yet, I can follow the rationale in the Dockerfile, Snakefile and FUSTr wrapper and I do not think there should be any problem to reproduce the analysis step by step.

In my test, the docker daemon fails while trying to "apt-get install -y wget git build-essential cmake unzip curl" (all of which are part of the any linux distribution) and exits with a non-zero code. When I tried tweaking the FUSTr wrapper to run the pipeline, I faced an "snakemake: error: unrecognized arguments: --use-conda" which leas to another non-zero exit status 2.

$ conda -V
conda 4.3.30
$ snakemake -v
3.13.3

My suggestion at this point is that the authors document all the programs needed to run the pipeline and offer an alternative customizable wrapper that the user can modify to point to the local installation of the programs (do the user really needs to re-install everything every time?).

Validity of the findings

nothing to add

Comments for the author

nothing to add

---

## Round 0.3 · accepted · Accept

All the concerns have been properly addressed.